# Relevance of Costovertebral Exarticulation of the First Rib in Neurogenic Thoracic Outlet Syndrome: A Retrospective Clinical Study

**DOI:** 10.3390/jpm13010144

**Published:** 2023-01-11

**Authors:** Franz Lassner, Michael Becker, Andreas Prescher

**Affiliations:** 1Pauwelsklinik, Boxgraben 56, 52064 Aachen, Germany; 2Institute für Molecular und Cellular Anatomy, Aachen University, Pauwelsstr. 30, 52074 Aachen, Germany

**Keywords:** neurogenic thoracic outlet syndrome, nerve compression syndrome, costoclavicular exarticulation

## Abstract

Purpose: The failure rate for operative decompression in neurogenic thoracic outlet syndrome (NTOS) is high compared to more distal nerve compression syndromes, such as cubital or carpal tunnel syndrome. Herein, we aimed to determine if a more radical approach, namely costovertebral exarticulation of the first rib, may improve the postoperative results in patients with NTOS. Methods: From October 2002 to December 2020, 105 operative decompressions in 95 patients were evaluated; in 10 cases, decompressions were performed bilaterally. We presented the clinical outcomes of 59 exarticulations compared to those of 46 conventional resections. Evaluation was performed at a minimum of one year post-operation using the DASH questionnaire. Results: The exarticulation group presented with significantly better clinical outcomes (two-sample *t*-test assuming unequal variances, *p* < 0.001). Conclusions: This study showed that significantly better results were obtained when exarticulation of the first rib was performed in patients with NTOS. This finding supports the hypothesis that, in certain cases, the proximal portion of the first rib plays a pivotal role in the pathogenesis of NTOS.

## 1. Introduction

Thoracic outlet syndrome (TOS) is a compression syndrome of the upper extremity with neurologic, arterial, and venous manifestations. Most patients (90%) present with predominantly neurological symptoms (neurogenic TOS [NTOS]) [1]. Controversies exist regarding diagnosis and therapy. No imaging or neurophysiological methods are available for the diagnosis of TOS; therefore, diagnosis is primarily clinical [2,3,4]. Surgery is recommended for patients who do not respond to conservative therapy [5]. Operative decompression is technically demanding, with a potential risk of major complications and unfavourable results [6]; the rate of recurrence or therapy failure is reported to be 10–30% [2,7,8,9]).

The cause of nerve compression is postulated to be a spatial narrowness in the scalene triangle, whereas a forceps-like compression of the brachial plexus between the clavicle and the first rib is assumed to be one of the underlying pathologic mechanisms [1,3,10]. A hypertrophic scalenus medius muscle, long cervical ribs, or fibrous ligaments can intensify the pathology, as can malpositioning of the clavicle [3]. The first description of axillary access to resection the first rib was made in 1966 by Roos [11]. Surgical therapy now aims at an expansion of the costoclavicular space by means of scalenectomy, partial resection of the first rib and, if present, resection of any existing anatomical variants such as cervical ribs or fibrous ligaments. Authors who have studied larger numbers of cases have reported performing scalenectomy in combination with the partial resection of the first rib [12,13]. However, a substantial number of patients, with figures ranging between 10% and 30%, did not do well after the procedure; a number of causes are being considered to explain these failures, such as postoperative fibrous adhesions as a result of hematoma, infections, lymph fistulas or mechanical irritation caused by the remaining posterior stump of the first rib [3,8,14,15].

In this context, the question must be raised as to what can be considered sufficient decompression and whether this is achieved with the established surgical procedures. Certain awareness for this issue exists; for example, Leffert states that inadequate resection of the first rib may aggravate the symptoms and may make them worse than without surgery [10]. Figure 1 shows the resection line depicted by Roos in 1966 [11].

Resection with no more than 1 or 2 cm of rib remaining at the proximal stump has been considered adequate [10,12,13]; one exception is the procedure reported by Illig, who describes the removal of the first rib, which includes the articular section of the vertebrocostal joint [16]. As we will explain in this study, the proximal 1–2 cm of the first rib seems to be a critical area for the pathogenesis of the TOS. The neuroforamen of the root Th1 (Thoracal 1) lies dorsally below the first rib. Therefore, for a distance of approximately 1 cm, the Th1 root runs adjacent to the caudal aspect of the first rib before it unites with the root C8 (Cervical 8) at the anterior margin to form the inferior trunk. Figure 2a shows the situs from a cranial to caudal viewing angle.

In Figure 2b, the first rib has been removed and the site of compression of the root Th1 at the anterior margin of the first rib is marked with an arrow.

Figure 3 shows the positional relationship of C8, Th1 and the inferior trunk to the first rib and the scalenus medius muscle.

Functionally, the first rib acts as a hypomochlion for the root Th1, analogous to the situation of the ulnar nerve in the cubital tunnel. The pressure load of the root Th1 at the anterior margin of the first rib may further be increased if a hypertrophic scalenus medius muscle or a scalenus medius muscle with an insertion reaching far distally on the first rib forces the inferior trunk into a more cranial position. The first rib, particularly wide in the area of the collum and caput, has a similar effect. The crossing point of the root Th1 with the first rib lies within the first 1–2 cm of the post-foraminal course. If this critical area of the rib remains in situ, it is conceivable that sufficient decompression of the brachial plexus may not be achieved.

We presented a study comparing the clinical results of a series of patients whose first rib was exarticulated at the vertebrocostal joint with those of a second series of patients who underwent the procedure using the conventional technique.

## 2. Methods

The work was performed in concordance to the ethical standards of the WMA (World Medical Association) Declaration of Helsinki. In our institution, scalenectomy and partial resection of the first rib were performed by two surgeons using a supraclavicular approach since 2002. Following the considerations outlined above, from February 2013 onwards, the first author performed proximal exarticulation of the first rib in all cases with a suitable anatomy. Informed consent was obtained from all patients who underwent the extended resection procedure after explaining the possible benefits and potential risks. We retrospectively analysed the outcomes of 105 decompression procedures in 95 patients. Only primary cases were evaluated, and all revision cases and all patients with relevant concomitant pathologies were excluded. Only patients with a minimum follow-up using the DASH score (Disabilities of Arm, Shoulder and Hand) questionnaire were included. Statistical analysis was performed with a two-sample t- test assuming unequal variancies.

We used a classification system with five degrees of severity according to the clinical criteria (Table 1), the indication for surgical intervention was set for patients at stages 3 and 4. Stage 1 had no surgical indication. Stage 2 patients were treated conservatively with physiotherapy, as well as with anti-inflammatory and analgesic medications. Stage 2 patients were recommended to avoid great physical strain and, in most cases, permanent physical therapy was required.

### 2.1. Surgical Technique: Exarticulation

The supraclavicular approach was used in all patients. With the patient in the supine position, disinfection of the neck, shoulder and arm is performed, with the arm in a freely movable position. After dissection of the platysma, the supraclavicular sensory branches are mobilised and retracted. Detachment of the lateral head of the sternocleidomastoid muscle from its clavicular insertion is usually required. The omohyoid muscle is identified and retracted. At its anterior border, further dissection is carried out with extreme caution to avoid lymph leakage. The superior trunk is identified and, if present, the deep cervical artery is ligated. The medial and inferior trunks are identified. At this point, possible anatomic causes such as fibrous bands and accessory muscles can be identified and removed, thus allowing the surgeon to tailor the surgical procedure to the individual’s anatomy. We used a 6 mm silicon drain instead of a vessel loop for further manipulation of the nerves to minimise the risk of pressure damage. The phrenic nerve is identified on the anterior scalene muscle, and the muscle is left intact. The subclavian artery is gently retracted anteriorly; then, the medial scalene muscle is detached from its costal insertion and retracted. Staying close to the bone, the circumference of the rib is freed from the intercostal muscle insertion. After identifying the vertebrocostal joint and Th1 root, dissection of the caput costae is carried out with a Kerrison rongeur medially and ventrally to the root Th1; the posterior part of the vertebrocostal joint is mobilised. Distally, the first rib is divided caudally to the clavicle. The first rib is then mobilised, the remaining muscle insertions are dissected and the rib is removed. A dynamic test is now applied; a finger is inserted into the situs parallel to the nerve trunks while moving the arm into full abduction and full anterversion, thus verifying the absence of any remaining anatomic obstacle. Figure 4 demonstrates the postoperative computed tomography (CT) scan after exarticulation of the rib.

### 2.2. Surgical Technique: Conventional Resection

Similar to the technique detailed above, dissection is carried out until the nerves are slung up with a silicone drain. After mobilisation and partial resection of the scalenus medius muscle, resection of the first rib is performed, leaving a proximal stump of 1–2 cm.

With either surgical procedure, the tissue block containing the lymph vessels is sutured back in place and the sternocleidomastoid muscle is reinserted. The wound is closed over capillary drainage. Postoperatively, the arm is supported with a cuff and collar sling, and patients are instructed on passive mobilisation of the shoulder beginning from the 3rd postoperative day to prevent capsular contracture. For the postoperative course, we advocated for a slow, pain-adapted increase in physical load, which may take from 6 months up to 1 year. As a rule, the shoulder regains full range of motion at 2 weeks postoperatively.

## 3. Results

In this study, we evaluated a total of 105 surgical procedures on 95 patients, subdivided into two groups: Group 1 (exarticulation, 52 patients, 59 procedures) and Group 2 (conventional resection, 43 patients, 46 procedures). The groups were comparable in terms of age and gender distribution. The indication for the procedure was set at the severity level of 3 and 4 in both groups. The male-to-female ratio was 9:44 in group 1 (exarticulation) and 11:32 in group 2 (conventional). The mean age at the time of surgery was 37.0 (exarticulation) and 38.1 (conventional). On the DASH questionnaire, Group 1 (exarticulation) scored a mean value of 41.87, compared to 62.26 in Group 2 (conventional resection), which represents a significant difference (two sample *t*-test assuming unequal variances, *p* < 0.001).

No major complications occurred in these series of patients, such as hematoma needing operative revision, permanent nerve lesions or infections. Five transient nerve lesions were recorded: three Horner syndromes (one case in Group 1 and in two cases in Group 2), one long thoracic nerve and one phrenic nerve (both in Group 2). One case of pleural injury was recorded in Group 2, but drainage was not required. High bleeding risk due to venous malformations at the level of the inferior trunk in two patients and difficult anatomic constellation in two patients prohibited exarticulation; the surgery had to be completed using the conventional technique. These four patients did not improve postoperatively, one needed operative revision for neurolysis. Two patients in Group 1 (exarticulation) needed operative revision due to persistent symptoms. We found adhesions from perineural scarring, which were treated by neurolysis.

## 4. Discussion

Due to the lack of objective testing procedures, the diagnosis of TOS remains a clinical one, rendering the topic highly controversial. The pathomechanisms are poorly understood [17], with the first rib playing a central role [1,8]. Three anatomical constrictions are identified as potential causes of compression: the thoracic outlet, the interscalenic region and the costoclavicular space. The first rib limits each of these spaces due to its topographic relationship to all three anatomical constrictions. Resection of the first rib is considered necessary, since all possible causes are addressed with this procedure; however, it is not possible to classify the potential causes based on their valency [10]. With scalenectomy and partial resection of the first rib, decompression is achieved by widening the anatomical spaces caudally. This is currently the standard procedure for surgical decompression of TOS [10,12,13,17]. Whether a superior (supraclavicular) or inferior (axillary) approach is used depends on the surgeon’s experience or preference. A combined procedure is used if rib resection is performed from the axillary and a scalenectomy is also considered necessary. Tenotomy of the pectoralis minor muscle may be necessary in certain cases.

Unfavourable postoperative results in nerve compression syndromes in general and in TOS in particular are caused either by postoperative scarring or by inadequate decompression [10,18,19,20]. The role of scarring in the genesis of recurrent TOS has been described by several authors and coincides with our experience [9,10,19]. As long as the pathogenetic mechanisms are not accessible to objective diagnostics, the question of which operative procedure leads to adequate decompression can only be answered by approximation. Our results raise doubts as to whether this can be achieved in every case using the standard methods of scalenectomy and partial resection of the first rib. Leffert described the pathomechanism of TOS as involving caudalisation of the plexus, which causes the nerves and vessels to be pulled onto and pressed against the first rib [10]. From our point of view, in contrast to this, cranialisation of the plexus subjects the root Th1 to increased tension at the anterior margin of the first rib. In this scenario, certain predisposing factors are important: a constitutionally broad first rib, a hypertrophic scalenus medius muscle, the base of which may extend far distally on the first rib [10], and accessory ligaments, all of which are typical intraoperative findings. These structures exert direct pressure on the truncular level of the brachial plexus, analogous to the pathogenesis of CTS. Additionally, the root Th1 is subjected to increased traction; the first rib in this constellation serves as a hypomochlion, which is analogous to certain aspects of the pathogenesis of cubital tunnel syndrome. Direct pressure relief at the truncular level of the plexus can be achieved through scalenectomy and partial resection of the first rib. Furthermore, a certain degree of caudalisation of the plexus will result from scalenectomy, which then has the effect of reducing the tension at the root Th1. Whether this alone is sufficient to achieve symptom relief while the remaining stump of the first rib remains in situ with the root Th1 running around this stump is difficult to assess based on intraoperative findings. Illig provides a detailed description of his operative technique, where the resection line of the first rib extends posteriorly to the articular level [16]. Our study provided data to support this more radical approach to bone resection, and we showed that significantly better results were obtained when exarticulation of the first rib was performed. This supports our hypothesis that the proximal portion of the first rib in certain cases plays a pivotal role in the pathogenesis of TOS. However, the exarticulation of the proximal part of the first rib is technically complex and time-consuming. Based on the considerations given above, it should be noted that it may not always be necessary to disarticulate the first rib to achieve sufficient symptom relief. This is documented by good clinical results, both in our Group 2 patients and as reported by other authors [10,12]. However, as mentioned before, we are confronted with a comparatively high rate of therapy failure. Currently, there are no imaging procedures that detect cases where disarticulation of the first rib may be required. However, if one does not want to operate radically in every case, the necessity of a revision procedure must be considered. According to our experience, disarticulation of the first rib can only be performed with tolerable risk if there is no scarring due to previous surgery or trauma. Revision surgery for TOS is associated with a significantly higher risk of intraoperative complications and should be avoided; this is also the opinion of other authors [6,19]. As a consequence of these considerations, the recommendation should be made in the current situation to attempt, as far as technically possible, to disarticulate the first rib for decompression of the TOS, even if in some cases this means surgical overtherapy. In our experience, this led to a significant improvement in the results. The indication for surgery should be made in stage 3, when no permanent sensorimotor deficits are present. Postoperatively, a majority of patients are symptom-free in the absence of heavy physical load, and about 20% of patients are symptom-free without any restrictions. For the surgical treatment of TOS, this may lead to a change in the indicative balance toward a more generous indication. With appropriate awareness, one may recognise that several therapy-resistant symptoms (e.g., CTS, epicondylitis, cubital tunnel syndrome and shoulder arm syndrome) are caused by TOS. There is a growing realisation that TOS is underdiagnosed, and we can endorse the views of other authors who studied a larger number of cases [10,21].

In conclusion, our data support the more radical approach of bone resection, showing that significantly better results were obtained when exarticulation of the first rib was performed. This supports the hypothesis that the proximal portion of the first rib plays a pivotal role in the pathogenesis of NTOS in certain cases. In our clinical practice, we prefer the more radical approach because the conventional technique may not lead to sufficient decompression of the brachial plexus.

## Figures and Tables

**Figure 1 jpm-13-00144-f001:**
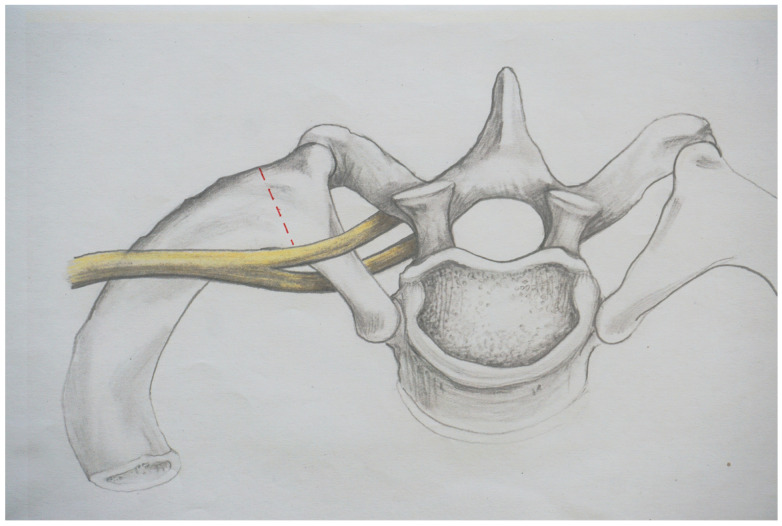
The resection line, as described by Roos in 1966 [15], is indicated by the dotted line.

**Figure 2 jpm-13-00144-f002:**
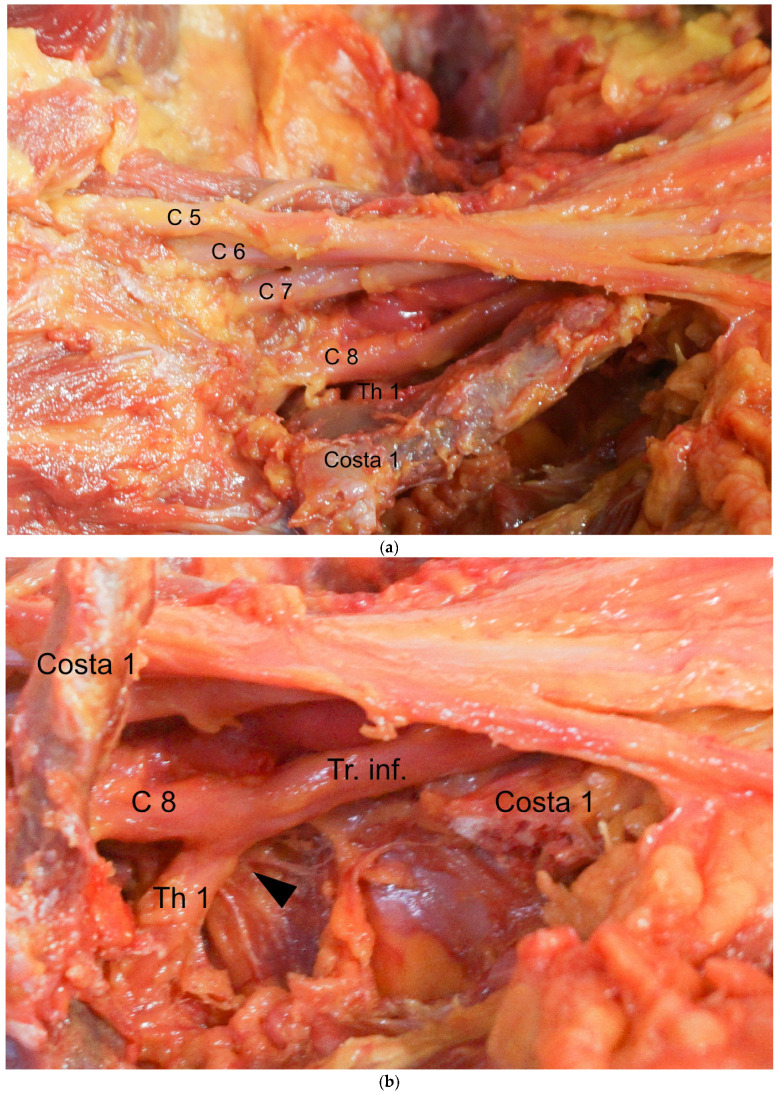
(**a**) The root Th1 at the anterior margin of the first rib is shown. (**b**) The first rib has been removed; the site of compression of the root Th1 at the anterior margin of the first rib is marked with an arrow.

**Figure 3 jpm-13-00144-f003:**
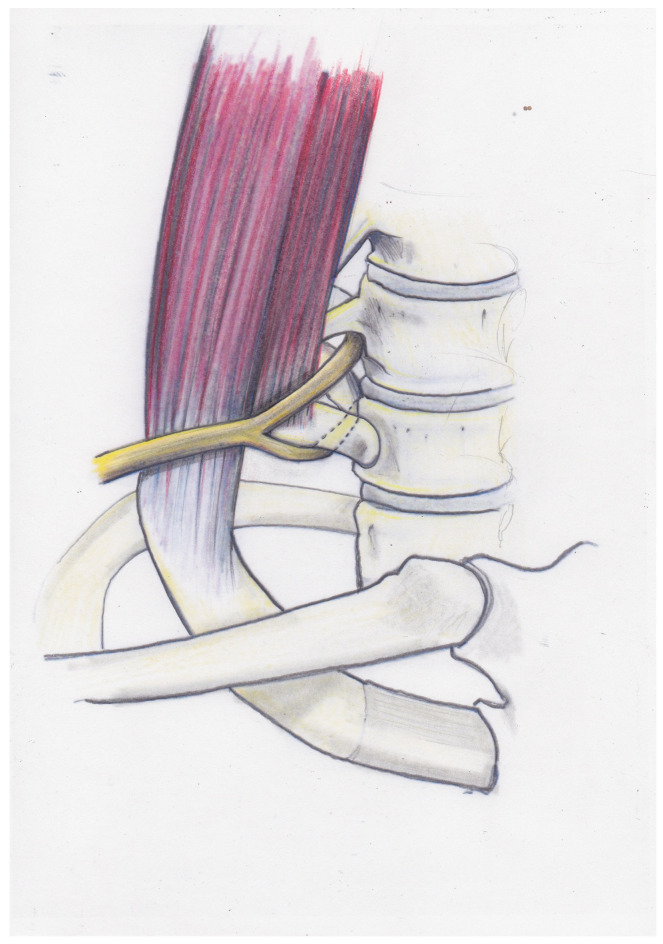
The positional relationship of C8, Th1 and the inferior trunk to the first rib and the scalenus medius muscle is demonstrated.

**Figure 4 jpm-13-00144-f004:**
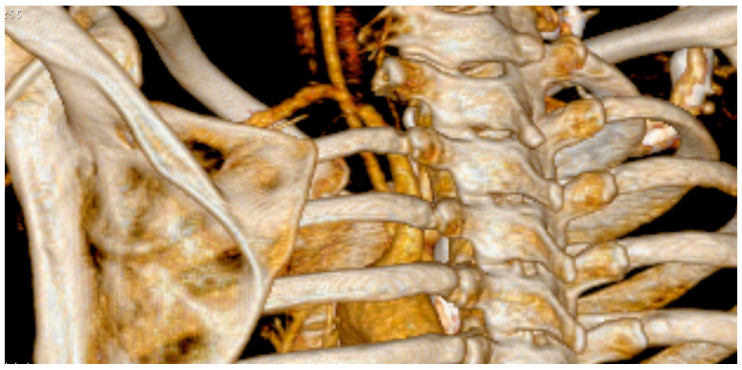
Postoperative CT scan after exarticulation of the left first rib.

**Table 1 jpm-13-00144-t001:** Thoracic Outlet Syndrome, classification.

Stage	Symptoms
0	None, no limitations for physical load.
1	Symptoms when submitted to severe physical load, with corresponding occupational limitations; symptom-free in daily activities.
2	Symptoms when submitted to moderate physical load, with limitations in daily activities.
3	Symptoms under light physical load, severe limitation in daily activities.
4	Permanent symptoms, motorsensory deficits.

## Data Availability

Not applicable.

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
