# Peer review of "Relevance of Costovertebral Exarticulation of the First Rib in Neurogenic Thoracic Outlet Syndrome: A Retrospective Clinical Study"

_jpm, 2023, doi:10.3390/jpm13010144_

Round 1
Reviewer 1 Report
Well written paper, with some limitations in the methodology
Line 101 fully write what th1 stands for as it is the first time it appears in the main text (same with “C”, “WMA”, “DASH”… further in the text)
Line 117-118 add bibliography
Line 129 it is not clear in Methods when the surgery was “technically possible”… explicit … and how patients were divided in 2 different groups
Line 186 in the results paragraph it could be interesting to discuss how many and which anatomic causes (fibrous band, accessory muscle…) were present in each group. Furthermore failure rates are not clear in this paragraph. Finally I believe the use of DASH as only measure to evaluate those 2 techniques is limiting.
Author Response
Unfortunately, line numbering in your text did not correspond to the numbering in the downloaded manuscript, but I have modified the text according to your suggestions.
The changes are highlighted in red.
Line 101, changes have been made in the manuscript
Line 117-118: Unfortunately, due to the incongruent numbering it is not evident to me where additional bibliographic information is necessary.
Line 129: “Technically possible” refers to anatomic situations, especially in obese patients, where the proximal part of the 1st rib was not accessible without exerting pathological traction to the inferior trunc and the root Th1. In certain cases varicous malformations at the level of the inferior trunc imposed an unacceptable risk of uncontrollable bleeding. Chances have been made in the text, also addressing the matter of subdividing the patients into two groups.
Line 186: Presence of anatomic causes:
Accessory muscles are a rare entity as far as our experience goes. In the majority of cases we encountered a hypertrophic medial scalene muscle with a long, sharp fibrous insertion to the first rib. This was treated by scalenotomy as specified in the manuscript. Fibrous bands as manifestation of a rudimentary cervical rib were present in 6 cases of group 1 and 4 cases of group 2. Resection of these fibrous bands was part of the surgical procedure during the process of approaching the first rib. Finally, as a dynamic test, we checked for any possible remaining anatomic obstacles before concluding the operation, as I have outlined in the revised manuscript.
Concerning the limitations of the DASH test, to our knowledge this is the first paper which adresses the results of TOS operations with a standardized test procedure, in contrast to evaluations along patient satisfaction or fair/good/excellent schemes as has been provided to date in other publications. We have collected data on pain pre- and post op, using the VAS scheme, but decided not to include these data due to the subjective bias.
Sincerely
F. Lassner
Reviewer 2 Report
I'd like to congratulate the authors for this very well written manuscript and the illustration of this highly surgical demanding technique. Please find my comments in the following paragraph below:
General questions:
Do you think that these cases should be ascribed to a new syndrome as it represents more of a proximal nerve root compression syndrome and less of a traditional text-book" nTOS pathology? Postoperative aggravated over-stretching of the Th1 nerve root may be the reason for cases with non resolved or even increased pain after conventional surgery.
Line 104: I'd recommend rephrasing this part: "from a cranial to caudal viewing angle."
Line 135: This part is rather confusing. "From February 2013 onwards, the first author performed proximal exarticulations of the first rib in all cases with a suitable anatomy. Informed consent was obtained from all patients prior to surgery.
Line 139: "Only patients with a minimum follow-up of 1 year were included."
Line 142: I'd recommend using the term "classification system."
Line 160: Minimise the risk of pressure damage.
Line 178: The term "replaced" may be misleading in this context. I'd use something like "sutured back in place."
Line 194: While acknowleding the vast score improvement, the numbers are still quite high. Why do you think that is assuming all anatomical areas of potential compression have been removed?
Author Response
Madam, Sir,
thank you for revising the manuscript, I appreciate the comments and have made changes accordingly.
Concerning you introductory question, I can see your point. However, I do not regard these cases as a new syndrome, the clinical appearance is well within the typical scenario of nTOS pathology, with its immanent variety of symptoms. However, the clinical appearance overlaps with the typical picture of root compression, and cervical spine pathology has to be considered as differential diagnosis, which is our policy and which is in accordance to the literature/textbooks. When in doubt, we want to have a cervical MRI prior to surgery enabling us to exclude relevant pathology of the cervical spine as a possible cause of the symptoms.
Postoperative aggravated over-stretching of the Th1 root is an issue regarding non responders, but it is my observation that this occurs to a much lesser extent in these patients where the more radical approach and been adopted. The majority of these patients will be able to elevate the operated arm on the first postop. day to an average of 140°, without exerting the typical neurologic symptoms. All patients are encouraged to elevate the operated arm once a day as far as pain permits, with gradually increasing ROM, most of the patients will gain 170° at the time of stich removal. Not all of them will manage this, there is always a great variety of individual pain perception.
Non responders are a wellknown issue to every surgeon who is engaged into TOS surgery. In our perception, this is caused either by insufficient decompression, the topic which we address with this manuscript. The other major cause is postoperative scarring, caused by operative trauma, lymph leakage or inflammation. Meticulous surgical technique is mandatory to prevent these complications, or to keep the rate to a minimum.
Unfortunately, the line numbering in your text did not correspond to the numbering in the downloaded manuscript, but I have modified the text according to your suggestions. With reference to line 194 in your system I have added a sentence addressing a dynamic intraoperative test which we always perform but have not mentioned in the manuscript.
Happy New Year
F. Lassner